# A Convex Duality Framework for GANs

**Farzan Farnia**\*
farnia@stanford.edu

**David Tse**\*
dntse@stanford.edu

## Abstract

Generative adversarial network (GAN) is a minimax game between a generator mimicking the true model and a discriminator distinguishing the samples produced by the generator from the real training samples. Given an unconstrained discriminator able to approximate any function, this game reduces to finding the generative model minimizing a divergence score, e.g. the Jensen-Shannon (JS) divergence, to the data distribution. However, in practice the discriminator is constrained to be in a smaller class $\mathcal{F}$ such as convolutional neural nets. Then, a natural question is how the divergence minimization interpretation will change as we constrain $\mathcal{F}$. In this work, we address this question by developing a convex duality framework for analyzing GAN minimax problems. For a convex set $\mathcal{F}$, this duality framework interprets the original vanilla GAN problem as finding the generative model with the minimum JS-divergence to the distributions penalized to match the moments of the data distribution, with the moments specified by the discriminators in $\mathcal{F}$. We show that this interpretation more generally holds for f-GAN and Wasserstein GAN. We further apply the convex duality framework to explain why regularizing the discriminator's Lipschitz constant, e.g. via spectral normalization or gradient penalty, can greatly improve the training performance in a general f-GAN problem including the vanilla GAN formulation. We prove that Lipschitz regularization can be interpreted as convolving the original divergence score with the first-order Wasserstein distance, which results in a continuously-behaving target divergence measure. We numerically explore the power of Lipschitz regularization for improving the continuity behavior and training performance in GAN problems.

## 1 Introduction

Learning a probability model from data samples is a fundamental task in unsupervised learning. The recently developed generative adversarial network (GAN) [1] leverages the power of deep neural networks to successfully address this task across various domains [2]. In contrast to traditional methods of parameter fitting like maximum likelihood estimation, the GAN approach views the problem as a *game* between a *generator $G$* whose goal is to generate fake samples that are close to the real data training samples and a *discriminator $D$* whose goal is to distinguish between the real and fake samples. The generator creates the fake samples by mapping from random noise input.

The following minimax problem is the original GAN problem, also called *vanilla GAN*, introduced in [1]

$$\min_{G \in \mathcal{G}} \max_{D \in \mathcal{F}} \mathbb{E}\big[\log D(\mathbf{X})\big] + \mathbb{E}\big[\log\big(1 - D(G(\mathbf{Z}))\big)\big]. \tag{1}$$

Here $\mathbf{Z}$ denotes the generator's noise input, $\mathbf{X}$ represents the random vector for the real data distributed as $P_{\mathbf{X}}$, and $\mathcal{G}$ and $\mathcal{F}$ respectively represent the generator and discriminator function sets. Implementing this minimax game using deep neural network classes $\mathcal{G}$ and $\mathcal{F}$ has lead to the state-of-the-art generative model for many different tasks.

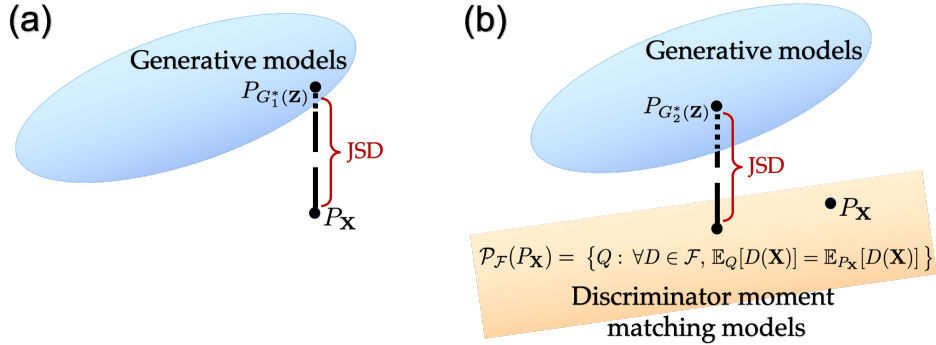

Figure 1: (a) Divergence minimization in vanilla GAN with D unconstrained, between the generative models and $P_{\mathbf{X}}$, (b) Divergence minimization in vanilla GAN with D constrained to a linear space $\mathcal{F}$, between the generative models and the discriminator moment matching models formed around $P_{\mathbf{X}}$.

To shed light on the probabilistic meaning of vanilla GAN, [1] shows that given an unconstrained discriminator $D$, i.e. if $\mathcal{F}$ contains all possible functions, the minimax problem (1) will reduce to

$$\min_{G \in \mathcal{G}} \mathrm{JSD}(P_{\mathbf{X}}, P_{G(\mathbf{z})}), \qquad (2)$$

where JSD denotes the Jensen-Shannon (JS) divergence. The optimization problem (2) can be interpreted as finding the closest generative model to the data distribution $P_{\mathbf{X}}$ (Figure 1a), where distance is measured using the JS-divergence. Various GAN formulations were later proposed by changing the divergence measure in (2). f-GAN [3] generalizes vanilla GAN by minimizing a general f-divergence. Wasserstein GAN (WGAN) [4] is based on the first-order Wasserstein (the earth-mover's) distance. MMD-GAN [5, 6, 7] considers the maximum mean discrepancy. Energy-based GAN [8] uses the total variation distance. Quadratic GAN [9] finds the distribution minimizing the second-order Wasserstein distance.

However, GANs trained in practice differ from this minimum divergence formulation, since their discriminator is not optimized over an unconstrained set and is constrained to smaller classes such as convolutional neural nets. As shown in [9, 10], constraining the discriminator is in fact necessary to guarantee good generalization properties for a GAN's learned model. Then, how does the minimum divergence interpretation illustarted in Figure 1a change after we constrain the discrminator? An existing approach used in [10, 11] is to view the maximum discriminator objective as a discriminator class $\mathcal{F}$-based distance between probability distributions. For unconstrained $\mathcal{F}$, the $\mathcal{F}$-based distance reduces to the original divergence measure, e.g. the JS-divergence in vanilla GAN.

While [10] demonstrates a useful application of $\mathcal{F}$-based distances in analyzing GANs' generalization properties, the connection between $\mathcal{F}$-based distances and the original divergence score remains unclear for a constrained $\mathcal{F}$. Then, what is the probabilistic interpretation of GAN minimax game in practice where a constrained discriminator is used? In this work, we address this question by interpreting the dual problem to the discriminator maximization problem. To analyze the dual problem, we develop a convex duality framework for divergence minimization problems with generalized moment matching constraints. We apply this convex duality framework to the f-divergence and Wasserstein distance families, providing interpretation for f-GAN, including vanilla GAN minimizing the JS-divergence, and Wasserstein GAN.

Specifically, we generalize [1]'s interpretation of the vanilla GAN problem (1), which only holds for an unconstrained discriminator set, to the more general case with linear space discriminator sets. Under this assumption, we interpret vanilla GAN as the following JS-divergence minimization between two sets of probability distributions (Figure 1b), the generative models and the discriminator moment-matching models,

$$\min_{G \in \mathcal{G}} \min_{Q \in \mathcal{P}_{\mathcal{F}}(P_{\mathbf{X}})} \mathrm{JSD}(P_{G(\mathbf{z})}, Q). \qquad (3)$$

Here $\mathcal{P}_{\mathcal{F}}(P_{\mathbf{X}})$ denotes the set of discriminator moment matching models that contains any distribution $Q$ satisfying moment matching constraints $\mathbb{E}_Q[D(\mathbf{X})] = \mathbb{E}_P[D(\mathbf{X})]$ for any discriminator $D \in \mathcal{F}$.

More generally, we show that a similar interpretation holds for GANs trained over convex discriminator sets. We also discuss the application of our duality framework to neural net discriminators with bounded Lipschitz constants. While a set of neural network functions is not necessarily convex, we prove that a convex combination of Lipschitz-bounded neural nets can be approximated by uniformly combining boundedly-many neural net functions. This result applied to our duality framework shows that the convex duality interpretation approximately holds for neural net discriminators.

As a byproduct, we apply the duality framework to the infimal convolution hybrid of f-divergence and the first-order Wasserstein ($W_1$) distance, e.g. the following hybrid of JS-divergence and $W_1$ distance:

$$d_{\text{JSD},W_1}(P_1, P_2) := \min_Q \ W_1(P_1, Q) + \text{JSD}(Q, P_2). \tag{4}$$

We prove that unlike the JS-divergence this hybrid divergence changes continuously and remedies the undesired discontinuous behavior of JS-divergence in optimizing generator parameters for vanilla GAN. [4] observes this issue with minimzing the JS-divergence in vanilla GAN and proposes to instead minimize the continuously-changing $W_1$ distance in WGAN. However, as empirically demonstrated in [12] vanilla GAN with a Lipschitz-bounded discriminator results in superior and state-of-the-art generative models over multiple benchmark tasks. In this paper, we leverage the convex duality framework to prove that the infimal convolution hybrid $d_{\text{JSD},W_1}$, possessing the same desired continuity property as in the $W_1$-distance, is in fact the divergence score minimized in vanilla GAN with a Lipschitz-bounded discriminator. Hence, our analysis provides an explanation for why regularizing the discriminator's Lipschitz constant via gradient penalty [13] or spectral normalization [12] greatly improves the training performance in vanilla GAN. We then extend our focus to the infimal convolution hybrid between the f-divergence and the second-order Wasserstein ($W_2$) distance. In this case, we derive the f-GAN (e.g. vanilla GAN) problem with its discriminator being adversarially trained over the generator's samples. We numerically evaluate the power of these hybrid divergences and their implied regularization schemes for training GANs.

## 2 Divergence Measures

### 2.1 Jensen-Shannon divergence

The Jensen-Shannon divergence is defined in terms of the KL-divergence (denoted by KL) as

$$\text{JSD}(P, Q) := \frac{1}{2} \text{KL}(P \| M) + \frac{1}{2} \text{KL}(Q \| M)$$

where $M = \frac{P+Q}{2}$ is the mid-distribution between $P$ and $Q$. Unlike the KL-divergence, the JS-divergence is symmetric $\text{JSD}(P, Q) = \text{JSD}(Q, P)$ and bounded $0 \leq \text{JSD}(P, Q) \leq \log 2$.

### 2.2 f-divergence

The f-divergence family [14] generalizes the KL and JS divergence measures. Given a convex lower semicontinuous function $f$ with $f(1) = 0$, the f-divergence $d_f$ is defined as

$$d_f(P, Q) := \mathbb{E}_P\big[f\big(\frac{q(\mathbf{X})}{p(\mathbf{X})}\big)\big] = \int p(\mathbf{x}) f\big(\frac{q(\mathbf{x})}{p(\mathbf{x})}\big) \, d\mathbf{x}. \tag{5}$$

Here $\mathbb{E}_P$ denotes expectation over distribution $P$ and $p, \ q$ denote the density functions for distributions $P, \ Q$, respectively. The KL-divergence and the JS-divergence are members of the f-divergence family, corresponding to respectively $f_{\text{KL}}(t) = t \log t$ and $f_{\text{JSD}}(t) = \frac{t}{2} \log t - \frac{t+1}{2} \log \frac{t+1}{2}$.

### 2.3 Optimal transport cost, Wasserstein distance

The optimal transport cost for cost function $c(\mathbf{x}, \mathbf{x}')$, which we denote by $W_c$, is defined as

$$W_c(P, Q) := \inf_{M \in \Pi(P,Q)} \mathbb{E}\big[c(\mathbf{X}, \mathbf{X}')\big], \tag{6}$$

where $\Pi(P, Q)$ contains all couplings with marginals $P, Q$. The Kantorovich duality [15] shows that for a non-negative lower semi-continuous cost $c$,

$$W_c(P, Q) = \max_{D \text{ c-concave}} \mathbb{E}_P\big[D(\mathbf{X})\big] - \mathbb{E}_Q\big[D^c(\mathbf{X})\big], \tag{7}$$

where we use $D^c$ to denote $D$'s c-transform defined as $D^c(\mathbf{x}) := \sup_{\mathbf{x}'} D(\mathbf{x}') - c(\mathbf{x}, \mathbf{x}')$ and call $D$ c-concave if $D$ is the c-transform of a valid function. An important special case is the first-order Wasserstein ($W_1$) distance corresponding to the norm cost $c(\mathbf{x}, \mathbf{x}') = \|\mathbf{x} - \mathbf{x}'\|$, i.e.

$$W_1(P, Q) := \inf_{M \in \Pi(P,Q)} \mathbb{E}\big[\|\mathbf{X} - \mathbf{X}'\|\big]. \tag{8}$$

For the norm cost function, a function $D$ is c-concave if and only if $D$ is 1-Lipschitz, and the c-transform $D^c = D$ holds for a 1-Lipschitz $D$. Therefore, the Kantorovich duality result (7) implies

$$W_1(P, Q) = \max_{D \text{ 1-Lipschitz}} \mathbb{E}_P\big[D(\mathbf{X})\big] - \mathbb{E}_Q\big[D(\mathbf{X})\big]. \tag{9}$$

In this paper, we also consider and analyze the second-order Wasserstein ($W_2$) distance, corresponding to the norm-squared cost $c(\mathbf{x}, \mathbf{x}') = \|\mathbf{x} - \mathbf{x}'\|^2$, defined as

$$W_2(P, Q) := \inf_{M \in \Pi(P,Q)} \mathbb{E}\big[\|\mathbf{X} - \mathbf{X}'\|^2\big]^{1/2}. \tag{10}$$

## 3 Divergence minimization in GANs: a convex duality framework

In this section, we develop a convex duality framework for analyzing divergence minimization problems conditioned to moment-matching constraints. Our framework generalizes the duality framework developed in [16] for the f-divergence family.

For a general divergence measure $d(P, Q)$, we define $d$'s convex conjugate for distribution $P$, which we denote by $d_P^*$, as the following operator mapping a real-valued function with domain $\mathcal{X}$ to a real number

$$d_P^*(D) := \sup_Q \mathbb{E}_Q[D(\mathbf{X})] - d(P, Q). \tag{11}$$

Here the supremum is over all distributions on the support set $\mathcal{X}$. The following theorem connects this operation to divergence minimization problems under moment matching constraints. Next section, we discuss the application of this theorem in deriving several well-known GAN formulations for divergence measures discussed in Section 2.

**Theorem 1.** *Suppose divergence $d(P, Q)$ is non-negative, lower semicontinuous and convex in distribution $Q$. Consider a convex set of continuous functions $\mathcal{F}$ and assume support set $\mathcal{X}$ is compact. Then,*

$$\min_{G \in \mathcal{G}} \max_{D \in \mathcal{F}} \mathbb{E}_{P_\mathbf{X}}[D(\mathbf{X})] - d_{P_{G(\mathbf{z})}}^*(D) \tag{12}$$
$$= \min_{G \in \mathcal{G}} \min_Q \big\{ d(P_{G(\mathbf{Z})}, Q) + \max_{D \in \mathcal{F}}\big\{ \mathbb{E}_{P_\mathbf{X}}[D(\mathbf{X})] - \mathbb{E}_Q[D(\mathbf{X})]\big\}\big\}.$$

*Proof.* We defer the proof to the Appendix. $\qquad\square$

Theorem 1 interprets the LHS minimax problem in (12) as finding the closest generative model to a set of distributions penalized to share the same generalized moments specified by discriminators in $\mathcal{F}$ with $P_\mathbf{X}$. The following corollary of Theorem 1 shows if we further assume that $\mathcal{F}$ is a linear space, then the additive penalty term penalizing the worst-case moment mismatch will turn to hard constraints in the discriminator optimization problem. This result reveals a divergence minimization problem between the generative models and the following set $\mathcal{P}_\mathcal{F}(P)$ which we call the discriminator moment matching models,

$$\mathcal{P}_\mathcal{F}(P) := \big\{ Q : \forall D \in \mathcal{F}, \mathbb{E}_Q[D(\mathbf{X})] = \mathbb{E}_P[D(\mathbf{X})] \big\}. \tag{13}$$

**Corollary 1.** *In Theorem 1 suppose $\mathcal{F}$ is also a linear space, i.e. for any $D_1, D_2 \in \mathcal{F}$, $\lambda \in \mathbb{R}$ we have $D_1 + D_2 \in \mathcal{F}$ and $\lambda D_1 \in \mathcal{F}$. Then,*

$$\min_{G \in \mathcal{G}} \max_{D \in \mathcal{F}} \mathbb{E}_{P_\mathbf{X}}[D(\mathbf{X})] - d_{P_{G(\mathbf{z})}}^*(D) = \min_{G \in \mathcal{G}} \min_{Q \in \mathcal{P}_\mathcal{F}(P_\mathbf{X})} d(P_{G(\mathbf{z})}, Q). \tag{14}$$

In next section, we apply this duality framework to divergence measures discussed in Section 2 and show how to derive various GAN problems through this convex duality framework.

# 4 Duality framework applied to different divergence measures

## 4.1 f-divergence: f-GAN and vanilla GAN

Theorem 2 shows the application of Theorem 1 to an f-divergence. Here we use $f^*$ to denote $f$'s convex-conjugate [17], defined as $f^*(u) := \sup_t ut - f(t)$. Theorem 2 applies to a general f-divergence $d_f$ as long as the convex-conjugate $f^*$ is a non-deacreasing function, a condition met by all f-divergence examples discussed in [3] with the only exception of Pearson $\chi^2$-divergence.

**Theorem 2.** *Consider f-divergence $d_f$ where the corresponding $f$ has a non-decreasing convex-conjugate $f^*$. In addition to the assumptions in Theorem 1, suppose that $\mathcal{F}$ is closed to adding constants, i.e. $D + \lambda \in \mathcal{F}$ for any $D \in \mathcal{F}, \lambda \in \mathbb{R}$. Then, the minimax problem in the LHS of* (12) *and* (14)*, reduces to*

$$\min_{G \in \mathcal{G}} \max_{D \in \mathcal{F}} \mathbb{E}[D(\mathbf{X})] - \mathbb{E}\big[f^*\big(D(G(\mathbf{Z}))\big)\big]. \tag{15}$$

*Proof.* We defer the proof to the Appendix. □

The minimax problem (15) is the f-GAN problem introduced and discussed in [3]. Therefore, Theorem 2 reveals that f-GAN searches for the generative model minimizing the f-divergence to the discriminator moment matching models specified by discriminator set $\mathcal{F}$. The following example shows the application of this result to the vanilla GAN introduced in the original GAN work [1].

**Example 1.** *Consider the JS-divergence, i.e. f-divergence corresponding to $f_{\mathrm{JSD}}(t) = \frac{t}{2} \log t - \frac{t+1}{2} \log \frac{t+1}{2}$. Then,* (15) *up to additive and multiplicative constants reduces to*

$$\min_{G \in \mathcal{G}} \max_{D \in \mathcal{F}} \mathbb{E}[D(\mathbf{X})] + \mathbb{E}\big[\log\big(1 - \exp(D(G(\mathbf{Z})))\big)\big]. \tag{16}$$

*Moreover, if for function set $\tilde{\mathcal{F}}$ the corresponding $\mathcal{F} = \{D : D(\mathbf{x}) = -\log(1 + \exp(\tilde{D}(\mathbf{x}))), \tilde{D} \in \tilde{\mathcal{F}}\}$ is a convex set, then* (16) *will reduce to the following minimax game which is the vanilla GAN problem* (1) *with sigmoid activation applied to the discriminator output,*

$$\min_{G \in \mathcal{G}} \max_{\tilde{D} \in \tilde{\mathcal{F}}} \mathbb{E}\big[ \log \frac{1}{1 + \exp(\tilde{D}(\mathbf{X}))} \big] + \mathbb{E}\big[ \log \frac{\exp(\tilde{D}(\mathbf{X}))}{1 + \exp(\tilde{D}(\mathbf{X}))} \big]. \tag{17}$$

## 4.2 Optimal Transport Cost: Wasserstein GAN

**Theorem 3.** *Let divergence $d$ be optimal transport cost $W_c$ where $c$ is a non-negative lower semi-continuous cost function. Then, the minimax problem in the LHS of* (12) *and* (14) *reduces to*

$$\min_{G \in \mathcal{G}} \max_{D \in \mathcal{F}} \mathbb{E}[D(\mathbf{X})] - \mathbb{E}\big[D^c(G(\mathbf{Z}))\big]. \tag{18}$$

*Proof.* We defer the proof to the Appendix. □

Therefore the minimax game between $G$ and $D$ in (18) can be viewed as minimizing the optimal transport cost between generative models and the distributions matching moments over $\mathcal{F}$ with $P_{\mathbf{X}}$'s moments. The following example applies this result to the first-order Wasserstein distance and recovers the WGAN problem [4] with a constrained 1-Lipschitz discriminator.

**Example 2.** *Let the optimal transport cost in* (18) *be the $W_1$ distance, and suppose $\mathcal{F}$ is a convex subset of 1-Lipschitz functions. Then, the minimax problem* (18) *will reduce to*

$$\min_{G \in \mathcal{G}} \max_{D \in \mathcal{F}} \mathbb{E}[D(\mathbf{X})] - \mathbb{E}\big[D(G(\mathbf{Z}))\big]. \tag{19}$$

Therefore, the moment-matching interpretation also holds for WGAN: for a convex set $\mathcal{F}$ of 1-Lipschitz functions WGAN finds the generative model with minimum $W_1$ distance to the distributions penalized to share the same moments over $\mathcal{F}$ with the data distribution. We discuss two more examples in the Appendix: 1) for the indicator cost $c_I(\mathbf{x}, \mathbf{x}') = \mathbb{I}(\mathbf{x} \neq \mathbf{x}')$ corresponding to the total variation distance we draw the connection to the energy-based GAN [8], 2) for the second-order cost $c_2(\mathbf{x}, \mathbf{x}') = \|\mathbf{x} - \mathbf{x}'\|^2$ we recover [9]'s quadratic GAN formulation under the LQG setting assumptions, i.e. linear generator, quadratic discriminator and Gaussian input data.

# 5 Duality framework applied to neural net discriminators

We applied the duality framework to analyze GAN problems with convex discriminator sets. However, a neural net set $\mathcal{F}_{nn} = \{f_{\mathbf{w}} : \mathbf{w} \in \mathcal{W}\}$, where $f_{\mathbf{w}}$ denotes a neural net function with fixed architecture and weights $\mathbf{w}$ in feasible set $\mathcal{W}$, does not generally satisfy this convexity assumption. Note that a linear combination of several neural net functions in $\mathcal{F}_{nn}$ may not remain in $\mathcal{F}_{nn}$.

Therefore, we apply the duality framework to $\mathcal{F}_{nn}$'s convex hull, which we denote by $\mathrm{conv}(\mathcal{F}_{nn})$, containing any convex combination of neural net functions in $\mathcal{F}_{nn}$. However, a convex combination of infinitely-many neural nets from $\mathcal{F}_{nn}$ is characterized by infinitely-many parameters, which makes optimizing the discriminator over $\mathrm{conv}(\mathcal{F}_{nn})$ computationally intractable. In the following theorem, we show that although a function in $\mathrm{conv}(\mathcal{F}_{nn})$ is a combination of infinitely-many neural nets, that function can be approximated by uniformly combining boundedly-many neural nets in $\mathcal{F}_{nn}$.

**Theorem 4.** *Suppose any function $f_{\mathbf{w}} \in \mathcal{F}_{nn}$ is $L$-Lipschitz and bounded as $|f_{\mathbf{w}}(\mathbf{x})| \le M$. Also, assume that the $k$-dimensional random input $\mathbf{X}$ is norm-bounded as $\|\mathbf{X}\|_2 \le R$. Then, any function in $\mathrm{conv}(\mathcal{F}_{nn})$ can be uniformly approximated over the ball $\|\mathbf{x}\|_2 \le R$ within $\epsilon$-error by a uniform combination $\hat{f}(\mathbf{x}) = \frac{1}{m}\sum_{i=1}^{m} f_{\mathbf{w}_i}(\mathbf{x})$ of $m = \mathcal{O}\big(\frac{M^2 k \log(LR/\epsilon)}{\epsilon^2}\big)$ functions $(f_{\mathbf{w}_i})_{i=1}^{m} \in \mathcal{F}_{nn}$.*

*Proof.* We defer the proof to the Appendix. $\qquad\square$

The above theorem suggests using a uniform combination of multiple discriminator nets to find a better approximation of the solution to the divergence minimization problem in Theorem 1 solved over $\mathrm{conv}(\mathcal{F}_{nn})$. Note that this approach is different from MIX-GAN [10] proposed for achieving equilibrium in GAN minimiax game. While our approach considers a uniform combination of multiple neural nets as the discriminator, MIX-GAN considers a randomized combination of the minimax game over multiple neural net discriminators and generators.

# 6 Infimal Convolution hybrid of f-divergence and Wasserstein distance: GAN with Lipschitz or adversarially-trained discriminator

Here we apply the convex duality framework to a novel class of divergence measures. For an f-divergence $d_f$, we define the divergence score $d_{f,W_1}$, which we call the infimal convolution hybrid of $d_f$ and $W_1$ divergence measures, as follows

$$d_{f,W_1}(P_1, P_2) := \inf_Q \; W_1(P_1, Q) + d_f(Q, P_2). \tag{20}$$

The above infimum is taken over all distributions on the support set $\mathcal{X}$, finding the distribution $Q^*$ minimizing the sum of the Wasserstein distance between $P_1$ and $Q$ and the f-divergence from $Q$ to $P_2$. Earlier in the introduction, we mentioned and discussed a special case of the above definition for the hybrid between the JS-divergence and $W_1$-distance. While f-divergence in f-GAN, e.g. JS-divergence in vanilla GAN, does not change continuously with the generator parameters, the following theorem proves that similar to the continuous behavior of $W_1$-distance shown in [18, 4] the infimal convolution hybrid divergence changes continuously with the generative model.

**Theorem 5.** *Suppose $G_{\boldsymbol{\theta}} \in \mathcal{G}$ is continuously changing with parameters $\boldsymbol{\theta}$. Then, for any $Q$ and $\mathbf{Z}$, $d_{f,W_1}(P_{G_{\boldsymbol{\theta}}(\mathbf{Z})}, Q)$ will behave continuously as a function of $\boldsymbol{\theta}$. Moreover, if $G_{\boldsymbol{\theta}}$ is assumed to be locally Lipschitz, then $d_{f,W_1}(P_{G_{\boldsymbol{\theta}}(\mathbf{Z})}, Q)$ will be differentiable w.r.t. $\boldsymbol{\theta}$ almost everywhere.*

*Proof.* We defer the proof to the Appendix. $\qquad\square$

Our next result reveals the minimax problem dual to minimizing this hybrid divergence with symmetric f-divergence component. We note that this symmetricity condition is met by the JS-divergence and the squared Hellinger divergence among the f-divergence examples discussed in [3].

**Theorem 6.** *Consider $d_{f,W_1}$ with a symmetric f-divergence $d_f$, i.e. $d_f(P,Q) = d_f(Q,P)$, satisfying the assumptions in Theorem 2. If the composition $f^* \circ D$ is 1-Lipschitz for all $D \in \mathcal{F}$, the minimax problem in Theorem 1 for the hybrid $d_{f,W_1}$ reduces to the f-GAN problem, i.e.*

$$\min_{G \in \mathcal{G}} \max_{D \in \mathcal{F}} \mathbb{E}[D(\mathbf{X})] - \mathbb{E}\big[ f^*\big(D(G(\mathbf{Z}))\big) \big]. \tag{21}$$

*Proof.* We defer the proof to the Appendix. $\qquad\square$

The above theorem reveals that if the discriminator's Lipschitz constant in f-GAN is properly regularized, then solving the f-GAN problem over the regularized discriminator in fact minimizes the continuously-changing divergence $d_{f,W_1}$. As a special case, in vanilla GAN (17) we only need to constrain the discriminator $\tilde{D}$ to be 1-Lipschitz, which can be done via the gradient penalty regularization [13] or the spectral normalization of $\tilde{D}$'s weight matrices [12]. Therefore, using these techniques we indeed minimize the continuously-behaving divergence score $d_{\mathrm{JSD},W_1}$. These results are consistent with [12]'s empirical results indicating that regularizing the discriminator's Lipschitz constant improves the training performance in vanilla GAN.

Our discussion has so far focused on convolving f-divergence with the first order Wasserstein distance, which translates into training f-GAN with a Lipschitz-bounded discriminator. As another solution, we show that the desired continuity property can also be achieved through the following infimal convolution with the second-order Wasserstein ($W_2$) distance-squared:

$$d_{f,W_2}(P_1,P_2) := \inf_Q \; W_2^2(P_1,Q) + d_f(Q,P_2). \tag{22}$$

**Theorem 7.** *Suppose $G_{\boldsymbol{\theta}} \in \mathcal{G}$ continuously changes with parameters $\boldsymbol{\theta} \in \mathbb{R}^k$. Then, for any distribution $Q$ and random vector $\mathbf{Z}$, $d_{f,W_2}(P_{G_{\boldsymbol{\theta}}(\mathbf{Z})}, Q)$ will be continuous in $\boldsymbol{\theta}$. Also, if we further assume $G_{\boldsymbol{\theta}}$ is bounded and locally-Lipschitz w.r.t. $\boldsymbol{\theta}$, then the divergence $d_{f,W_2}(P_{G_{\boldsymbol{\theta}}(\mathbf{Z})}, Q)$ is almost everywhere differentiable w.r.t. $\boldsymbol{\theta}$.*

*Proof.* We defer the proof to the Appendix. $\square$

The following result shows that minimizing $d_{f,W_2}$ reduces to f-GAN problem where the discriminator is being adversarially trained.

**Theorem 8.** *Assume $d_f$ and $\mathcal{F}$ satisfy the assumptions in Theorem 6. Then, the minimax problem in Theorem 1 corresponding to the hybrid $d_{f,W_2}$ divergence reduces to*

$$\min_{G \in \mathcal{G}} \max_{D \in \mathcal{F}} \mathbb{E}[D(\mathbf{X})] + \mathbb{E}\big[\min_{\mathbf{u}} -f^*\big(D(G(\mathbf{Z}) + \mathbf{u})\big) + \|\mathbf{u}\|^2\big]. \tag{23}$$

*Proof.* We defer the proof to the Appendix. $\square$

The above result reduces minimizing the hybrid $d_{f,W_2}$ divergence to an f-GAN minimax game with an additional third player. Here, the third player assists the generator by perturbing the generated fake samples in order to make them harder to be distinguished from the real samples by the discriminator. The cost for perturbing a fake sample $G(\mathbf{Z})$ to $G(\mathbf{Z}) + \mathbf{u}$ will be proportional to $\|\mathbf{u}\|^2$, constraining the power of the third player who plays adversarially against the discriminator. To implement the game between the three players, we can adversarially learn the discriminator while we are training GAN, via the Wasserstein risk minimization (WRM) adversarial learning scheme discussed in [19].

# 7  Numerical Experiments

To evaluate our theoretical results, we used the CelebA [20] and LSUN-bedroom [21] datasets. Furthermore, in the Appendix we include the results of our experiments over the MNIST [22] dataset. We considered vanilla GAN [1] with the minimax formulation in (17) and DCGAN [23] convolutional architecture for the neural net discriminator and generator. We used the code provided by [13] and trained DCGAN via Adam optimizer [24] for 200,000 generator iterations. We applied 5 discriminator updates per generator update.

Figure 2 shows how the discriminator loss evaluated over 2000 validation samples, which is an estimate of the divergence measure, changed as we trained the DCGAN over LSUN samples. Using standard DCGAN regularizied by only batch normalization (BN) [25], we observed (Figure 2- top left) that the JS-divergence estimate always remained close to its maximum value $\log_2 2 = 1$ and also correlated poorly with the visual quality of the generated samples. In this experiment, the vanilla GAN training failed and led to mode collapse starting at about the 110,000th iteration. On the other hand, after replacing BN with two different Lipschitz regularization tecniques, spectral normalization (SN) [12] and gradient penalty (GP) [13], to ensure that the discriminator is 1-Lipschitz, the discriminator loss decreased in a continuous monotonic fashion (Figures 2-top right and 2-bottom left).

These observations are consistent with Theorems 5 and 6 showing that the discriminator loss will become an estimate for the infimal convolution hybrid $d_{\mathrm{JSD},W_1}$ divergence which is behaving continuously with generator parameters. Also, the samples generated by the Lipschitz-regularized DCGAN

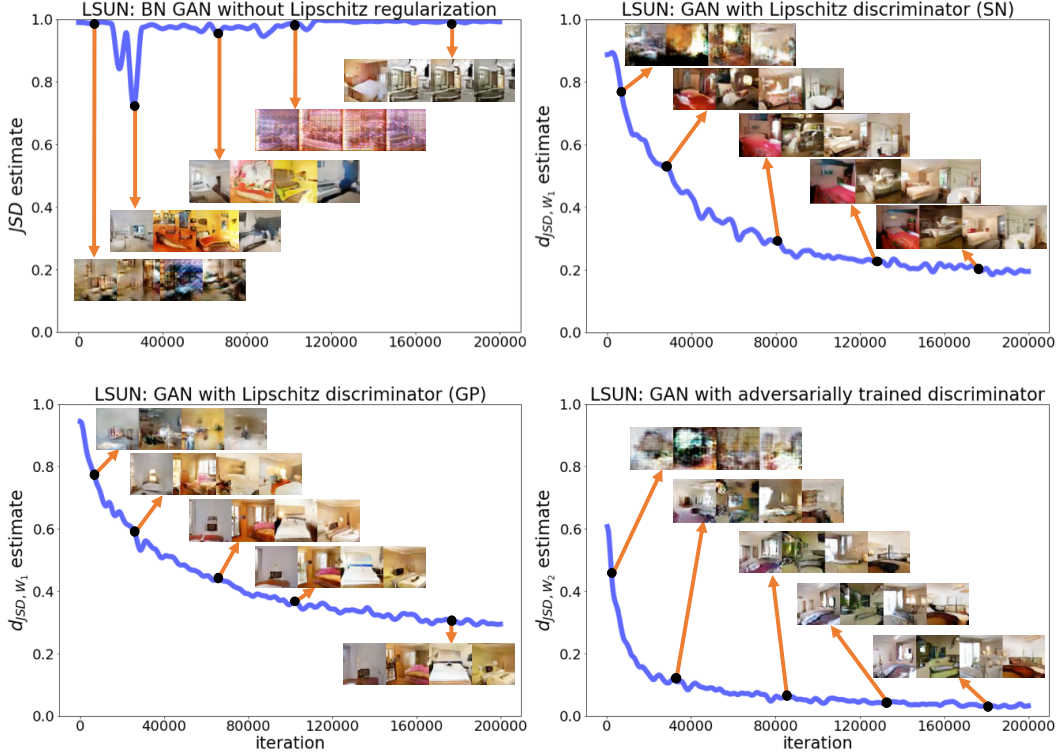

Figure 2: Divergence estimate in DCGAN trained over LSUN samples, (top-left) JS-divergence in DCGAN regularized with batch normalization (BN), (top-right) hybrid $d_{\text{JSD},W_1}$ in DCGAN with a 1-Lipschitz spectrally-normalized (SN) discriminator, (bottom-left) hybrid $d_{\text{JSD},W_1}$ in DCGAN with a 1-Lipschitz discriminator regularized via the gradient penalty (GP), (bottom-right) hybrid $d_{\text{JSD},W_2}$ in DCGAN with discriminator being adversarially-trained using WRM.

looked qualitatively better and correlated well with the estimate of $d_{\text{JSD},W_1}$ divergence. Figure 2-bottom right shows that a similar desired behavior with nice monotonic decrease in discriminator's loss can also be achieved through minimizing the second-order hybrid divergence $d_{\text{JSD},W_2}$. In this experiment, we trained the discriminator in vanilla GAN via the Wasserstein risk minimization (WRM) adversarial learning scheme [19].

Figure 3 shows the results of similar experiments over the CelebA dataset. Again, we observed (Figure 3-top left) that the JS-divergence estimate remains close to 1 while training DCGAN with BN. However, after applying two different Lipschitz regularization methods, SN and GP in Figures 3-top right and bottom left, we observed that the hybrid $d_{\text{JSD},W_1}$ changed nicely and monotonically, and correlated well with the quality of samples generated. Figure 3-bottom right shows that a similar desired behavior can also be obtained after minimizing the second-order infimal convolution hybrid $d_{\text{JSD},W_2}$ divergence. We defer the presentation of some random samples generated by the generators trained in these experiments to the Appendix.

## 8 Related Work

Theoretical studies of GAN have focused on three different aspects: approximation, generalization, and optimization. Regarding the approximation properties of GANs, [11] studies GANs' approximation power through a moment-matching approach. The authors view the maximized discriminator objective as an $\mathcal{F}$-based adversarial divergence, showing that the adversarial divergence between two distributions will be at its minimum value if the two distributions have the same generalized moments specified by $\mathcal{F}$. Our convex duality framework provides a dual interpretation for their results and draws the connection between the adversarial diveregnce and the original divergence scores. [26] studies the f-GAN problem through an information geometric approach and the connection between the Bregman divergence and the f-divergence.

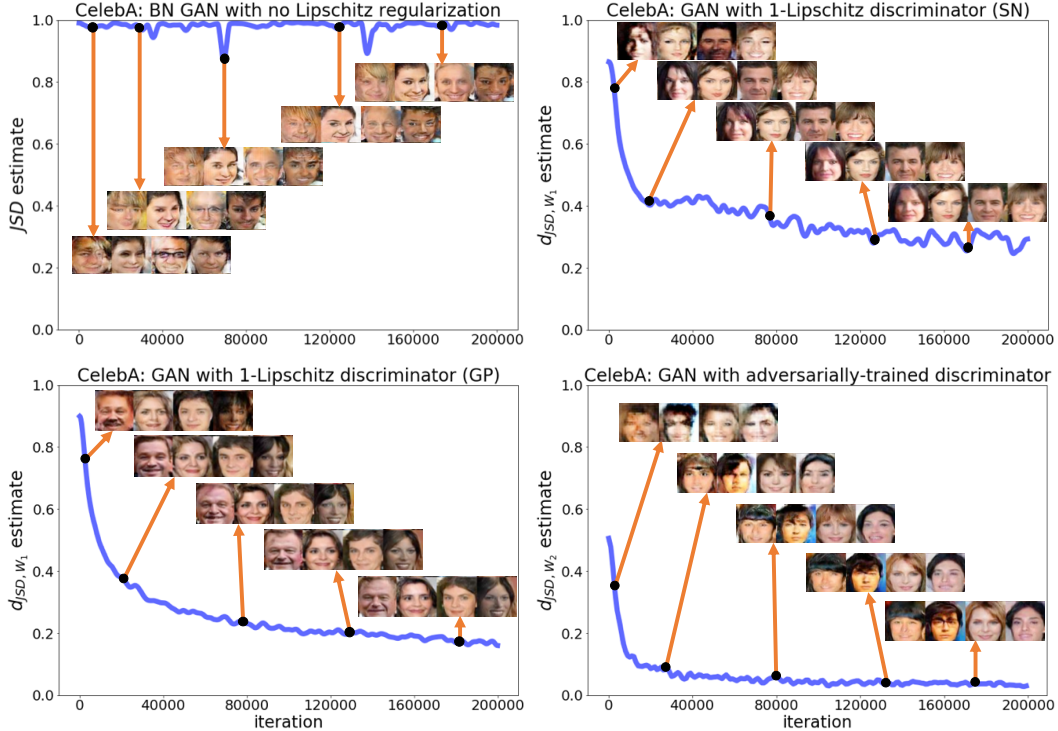

Figure 3: Divergence estimate in DCGAN trained over CelebA samples, (top-left) JS-divergence in DCGAN regularized with batch normalization, (top-right) hybrid $d_{\mathrm{JSD},W_1}$ in DCGAN with a 1-Lipschitz spectrally-normalized discriminator, (bottom-left) hybrid $d_{\mathrm{JSD},W_1}$ in DCGAN with a 1-Lipschitz discriminator regularized via the gradient penalty, (bottom-right) hybrid $d_{\mathrm{JSD},W_2}$ in DCGAN with its discriminator being adversarially-trained using WRM.

Analyzing the generalization performance in GANs has been another problem of interest in the machine learning literature. [10] proves generalization guarantee results for GANs in terms of the $\mathcal{F}$-based distance measures. [27] uses an elegant approach based on birthday paradox to empirically study the generalizibility of a GAN's learned models. [28] develops a quantitative approach for examining diversity and generalization for a GAN's learned distribution. [29] studies approximation-generalization trade-offs in GANs by analyzing the discriminative power in $\mathcal{F}$-based distances.

Regarding the optimization aspects of GANs, [30, 31] propose duality-based methods for improving optimization performance in training deep generative models. [32] suggests convolving the data distribution with a Gassian distribution for regularizing the learning problem in f-GANs. Moreover, several other works including [33, 34, 35, 9, 36] explore the optimization and stability properties of GANs. We also note that the same convex analysis approach used in this paper for studying GANs has also provided several powerful frameworks for analyzing other supervised and unsupervised learning problems [37, 38, 39, 40, 41].

**Acknowledgments:** We are grateful for support under a Stanford Graduate Fellowship, the National Science Foundation grant under CCF-1563098, and the Center for Science of Information (CSoI), an NSF Science and Technology Center under grant agreement CCF-0939370.

## Footnotes

\*Department of Electrical Engineering, Stanford University.

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
