[Supplementary Material]

# Appendix: A Convex Duality Framework for GANs

**Farzan Farnia**[*]
farnia@stanford.edu

**David Tse**[*]
dntse@stanford.edu

## 1 CelebA, LSUN, MNIST images generated by DCGANs trained via different methods

Figures 1, 2, and 3 show the CelebA, LSUN, and MNIST samples generated by the vanilla DCGAN trained via the methods described in the main text. Observe that applying Lipschitz regularization and adversarial training results in significantly higher quality generated samples. We note that tight SN in these figures refers to [1]'s spectral normalization scheme for convolutional layers, which precisely normalizes a conv layer's spectral norm and hence guarantees the 1-Lipschitzness of a discriminator convolutional neural net. Note that for non-tight SN we use the approximate scheme for normalizing a convolutional layer's operator norm introduced in [2].

(a) GAN with no regularization

(b) BN GAN with no Lipschitz regularization

(c) GAN with GP regularization

(d) GAN with SN regularization

(e) GAN with tight SN regularization

(f) GAN with WRM-trained discriminator

Figure 1: Samples generated by DCGAN trained over CelebA samples

---

[*]Department of Electrical Engineering, Stanford University.

(a) GAN with no regularization
(b) BN GAN with no Lipschitz regularization

(c) GAN with GP regularization
(d) GAN with SN regularization

(e) GAN with tight SN regularization
(f) GAN with WRM-trained discriminator

Figure 2: Samples generated by DCGAN trained over LSUN-bedroom samples

(a) GAN with no regularization
(b) BN GAN with no Lipschitz regularization

(c) GAN with GP regularization
(d) GAN with SN regularization

(e) GAN with tight SN regularization
(f) GAN with WRM-trained discriminator

Figure 3: Samples generated by DCGAN trained over MNIST samples

## 2 Proof of Theorem 1

Theorem 1 and Corollary 1 directly result from the following two lemmas.

**Lemma 1.** *Suppose divergence $d(P, Q)$ is non-negative, lower semicontinuous and convex in distribution $Q$. Consider a convex subset of continuous functions $\mathcal{F}$ and assume support set $\mathcal{X}$ is compact. Then, the following duality holds for any pair of distributions $P_1, P_2$:*

$$\max_{D \in \mathcal{F}} \mathbb{E}_{P_2}[D(\mathbf{X})] - d^*_{P_1}(D) = \min_Q \left\{ d(P_1, Q) + \max_{D \in \mathcal{F}} \left\{ \mathbb{E}_{P_2}[D(\mathbf{X})] - \mathbb{E}_Q[D(\mathbf{X})] \right\} \right\}. \quad (1)$$

*Proof.* Note that

$$
\begin{aligned}
& \min_Q \left\{ d(P_1, Q) + \max_{D \in \mathcal{F}} \left\{ \mathbb{E}_{P_2}[D(\mathbf{X})] - \mathbb{E}_Q[D(\mathbf{X})] \right\} \right\} \\
&= \min_Q \max_{D \in \mathcal{F}} \left\{ d(P_1, Q) + \mathbb{E}_{P_2}[D(\mathbf{X})] - \mathbb{E}_Q[D(\mathbf{X})] \right\} \\
&\overset{(a)}{=} \max_{D \in \mathcal{F}} \min_Q \left\{ d(P_1, Q) + \mathbb{E}_{P_2}[D(\mathbf{X})] - \mathbb{E}_Q[D(\mathbf{X})] \right\} \\
&= \max_{D \in \mathcal{F}} \left\{ \mathbb{E}_{P_2}[D(\mathbf{X})] + \min_Q \left\{ d(P_1, Q) - \mathbb{E}_Q[D(\mathbf{X})] \right\} \right\} \\
&= \max_{D \in \mathcal{F}} \left\{ \mathbb{E}_{P_2}[D(\mathbf{X})] - \max_Q \left\{ \mathbb{E}_Q[D(\mathbf{X})] - d(P_1, Q) \right\} \right\} \\
&\overset{(b)}{=} \max_{D \in \mathcal{F}} \mathbb{E}_{P_2}[D(\mathbf{X})] - d^*_{P_1}(D).
\end{aligned}
\quad (2)
$$

Here (a) is a consequence of the generalized Sion's minimax theorem [3], because the space of probability measures on compact $\mathcal{X}$ is convex and weakly compact [4], $\mathcal{F}$ is assumed to be convex, the minimiax objective is lower semicontinuous and convex in $Q$ and linear in $D$. (b) holds according to the conjugate $d^*_P$'s definition. $\qquad \square$

**Lemma 2.** *Assume divergence $d(P, Q)$ is non-negative, lower semicontinuous and convex in distribution $Q$ over compact $\mathcal{X}$. Consider a linear space subset of continuous functions $\mathcal{F}$. Then, the following duality holds for any pair of distributions $P_1, P_2$:*

$$\min_{Q \in \mathcal{P}_{\mathcal{F}}(P_2)} d(P_1, Q) = \max_{D \in \mathcal{F}} \mathbb{E}_{P_2}[D(\mathbf{X})] - d^*_{P_1}(D). \quad (3)$$

*Proof.* This lemma is a consequence of Lemma 1. Note that a linear space $\mathcal{F}$ is a convex set. Therefore, Lemma 1 applies to $\mathcal{F}$. However, since $\mathcal{F}$ is a linear space i.e. for any $D \in \mathcal{F}$ and $\lambda \in \mathbb{R}$ it includes $\lambda D$ we have

$$\max_{D \in \mathcal{F}} \left\{ \mathbb{E}_{P_2}[D(\mathbf{X})] - \mathbb{E}_Q[D(\mathbf{X})] \right\} = \begin{cases} 0 & \text{if } Q \in \mathcal{P}_{\mathcal{F}}(P_2) \\ +\infty & \text{otherwise.} \end{cases} \quad (4)$$

As a result, the minimizing $Q^*$ precisely matches the moments over $\mathcal{F}$ to $P_2$'s moments, which completes the proof. $\qquad \square$

## 3 Proof of Theorem 2

We first prove the following lemma.

**Lemma 3.** *Consider f-divergence $d_f$ corresponding to function $f$ which has a non-decreasing convex-conjugate $f^*$. Then, for any continuous $D$*

$$d_f{}^*_P(D) = \mathbb{E}_P\big[f^*\big(D(\mathbf{X}) + \lambda_0\big)\big] - \lambda_0 \quad (5)$$

*where $\lambda_0 \in \mathbb{R}$ satisfies $\mathbb{E}_P\big[f^{*\prime}\big(D(\mathbf{X}) + \lambda_0\big)\big] = 1$. Here $f^{*\prime}$ stands for the derivative of conjugate function $f^*$ which is supposed to be non-negative everywhere.*

*Proof.* Note that

$$
\begin{aligned}
d_f{}^*_P(D) &\overset{(a)}{=} \sup_Q \mathbb{E}_Q[D(\mathbf{X})] - d_f(P, Q) \\
&\overset{(b)}{=} \sup_Q \mathbb{E}_Q[D(\mathbf{X})] - \mathbb{E}_P\big[f\big(\frac{q(\mathbf{X})}{p(\mathbf{X})}\big)\big]
\end{aligned}
$$

$$\overset{(c)}{=} \max_{q(\mathbf{x})\geq 0,\, \int q(\mathbf{x})\,d\mathbf{x}=1} \int q(\mathbf{x})D(\mathbf{x})\,d\mathbf{x} - \mathbb{E}_P\Big[\,f\Big(\frac{q(\mathbf{X})}{p(\mathbf{X})}\Big)\Big]$$

$$\overset{(d)}{=} \min_{\lambda\in\mathbb{R}} -\lambda + \max_{q(\mathbf{x})\geq 0} \int q(\mathbf{x})\big(D(\mathbf{x})+\lambda\big)\,d\mathbf{x} - \mathbb{E}_P\Big[\,f\Big(\frac{q(\mathbf{X})}{p(\mathbf{X})}\Big)\Big]$$

$$\overset{(e)}{=} \min_{\lambda\in\mathbb{R}} -\lambda + \max_{r(\mathbf{x})\geq 0} \mathbb{E}_P\big[\,r(\mathbf{X})\big(D(\mathbf{X})+\lambda\big) - f(r(\mathbf{X}))\,\big]$$

$$\overset{(f)}{=} \min_{\lambda\in\mathbb{R}} -\lambda + \mathbb{E}_P\Big[\max_{r(\mathbf{X})\geq 0} r(\mathbf{X})\big(D(\mathbf{X})+\lambda\big) - f(r(\mathbf{X}))\Big]$$

$$\overset{(g)}{=} \min_{\lambda\in\mathbb{R}} -\lambda + \mathbb{E}_P\big[\,f^*\big(D(\mathbf{X})+\lambda\big)\,\big]$$

$$= -\max_{\lambda\in\mathbb{R}} \lambda - \mathbb{E}_P\big[\,f^*\big(D(\mathbf{X})+\lambda\big)\,\big] \tag{6}$$

$$\overset{(h)}{=} -\lambda_0 + \mathbb{E}_P\big[\,f^*\big(D(\mathbf{X})+\lambda_0\big)\,\big]. \tag{7}$$

Here (a) and (b) follow from the conjugate $d_P^*$ and f-divergence $d_f$ definitions. (c) rewrites the optimization problem in terms of the density function $q$ corresponding to distribution $Q$. (d) uses the strong convex duality to move the density constraint $\int q(\mathbf{x})\,d\mathbf{x}=1$ to the objective. Note that strong duality holds, since we have a convex optimization problem with affine constraints. (e) rewrites the problem after a change of variable $r(\mathbf{x})=q(\mathbf{x})/p(\mathbf{x})$. (f) holds since $f$ and $D$ are assumed to be continuous. (g) follows from the assumption that the derivative of $f^*$ takes non-negative values, and hence the minimizing $r(\mathbf{x})\geq 0$ also minimizes the unconstrained optimization for the convex conjugate $f^*$

$$f^*\big(D(\mathbf{X})+\lambda\big) := \max_{r(\mathbf{X})} r(\mathbf{X})\big(D(\mathbf{X})+\lambda\big) - f(r(\mathbf{X})).$$

Taking the derivative of the concave objective, the $\lambda$ value maximizing the objective solves the equation $\mathbb{E}_P\big[f^{*\prime}\big(D(\mathbf{X})+\lambda\big)\big]=1$ which is assumed to be $\lambda_0$. Therefore, (h) holds and the proof is complete. $\qquad\square$

Now we prove Theorem 2 which can be broken into two parts as follows.

**Theorem** (Theorem 2). *Consider f-divergence $d_f$ where $f$ has a non-decreasing conjugate $f^*$.*
*(a) Suppose $\mathcal{F}$ is a convex set closed to a constant addition, i.e. for any $D\in\mathcal{F}$, $\lambda\in\mathbb{R}$ we have $D+\lambda\in\mathcal{F}$. Then,*

$$\min_{P_{G(\mathbf{z})}\in\mathcal{P}_\mathcal{G}} \min_{Q_\mathbf{x}} d_f(P_{G(\mathbf{z})},Q) + \max_{D\in\mathcal{F}}\big\{\mathbb{E}_{P_\mathbf{x}}[D(\mathbf{X})] - \mathbb{E}_Q[D(\mathbf{X})]\big\}$$
$$= \min_{G\in\mathcal{G}} \max_{D\in\mathcal{F}} \mathbb{E}_{P_\mathbf{x}}[D(\mathbf{X})] - \mathbb{E}\big[f^*\big(D(G(\mathbf{Z}))\big)\big]. \tag{8}$$

*(b) Suppose $\mathcal{F}$ is a linear space including the constant function $D_0(\mathbf{x})=1$. Then,*

$$\min_{P_{G(\mathbf{z})}\in\mathcal{P}_\mathcal{G}} \min_{Q_\mathbf{x}\in\mathcal{P}_\mathcal{F}(P_\mathbf{x})} d_f(P_{G(\mathcal{Z})},Q) = \min_{G\in\mathcal{G}} \max_{D\in\mathcal{F}} \mathbb{E}_{P_\mathbf{x}}[D(\mathbf{X})] - \mathbb{E}\big[f^*\big(D(G(\mathbf{Z}))\big)\big]. \tag{9}$$

*Proof.* This theorem is an application of Theorem 1 and Corollary 1. For part (a) we have

$$\min_{P_{G(\mathbf{z})}\in\mathcal{P}_\mathcal{G}} \min_{Q_\mathbf{x}} d_f(P_{G(\mathbf{z})},Q) + \max_{D\in\mathcal{F}}\big\{\mathbb{E}_{P_\mathbf{x}}[D(\mathbf{X})] - \mathbb{E}_Q[D(\mathbf{X})]\big\}$$

$$\overset{(c)}{=} \min_{G\in\mathcal{G}} \max_{D\in\mathcal{F}} \mathbb{E}_{P_\mathbf{x}}[D(\mathbf{X})] - d_{f\,P_{G(\mathbf{z})}}^*(D)$$

$$\overset{(d)}{=} \min_{G\in\mathcal{G}} \max_{D\in\mathcal{F}} \mathbb{E}_{P_\mathbf{x}}[D(\mathbf{X})] + \max_{\lambda\in\mathbb{R}} \lambda - \mathbb{E}\big[f^*\big(D(G(\mathbf{Z}))+\lambda\big)\big]$$

$$= \min_{G\in\mathcal{G}} \max_{D\in\mathcal{F},\lambda\in\mathbb{R}} \mathbb{E}_{P_\mathbf{x}}[D(\mathbf{X})+\lambda] - \mathbb{E}\big[f^*\big(D(G(\mathbf{Z}))+\lambda\big)\big]$$

$$\overset{(e)}{=} \min_{G\in\mathcal{G}} \max_{D\in\mathcal{F}} \mathbb{E}_{P_\mathbf{x}}[D(\mathbf{X})] - \mathbb{E}\big[f^*\big(D(G(\mathbf{Z}))\big)\big].$$

Here (c) is a direct result of Theorem 1. (d) uses the simplified version (6) for $d_{f\,P}^*$. (e) follows from the assumption that $\mathcal{F}$ is closed to constant additions.

For part (b) note that since $\mathcal{F}$ is a linear space and includes $D_0(\mathbf{x}) = 1$, it is closed to constant additions. Hence, an application of Corollary 1 reveals

$$
\begin{aligned}
\min_{P_{G(\mathbf{z})} \in \mathcal{P}_{\mathcal{G}}} \min_{Q_{\mathbf{x}} \in \mathcal{P}_{\mathcal{F}}(P_{\mathbf{x}})} d_f(P_{G(\mathcal{Z})}, Q) &= \min_{G \in \mathcal{G}} \max_{D \in \mathcal{F}} \mathbb{E}_{P_{\mathbf{x}}}[D(\mathbf{X})] - d_f{}^*_{P_{G(\mathbf{z})}}(D) \\
&= \min_{G \in \mathcal{G}} \max_{D \in \mathcal{F}} \mathbb{E}_{P_{\mathbf{x}}}[D(\mathbf{X})] + \max_{\lambda \in \mathbb{R}} \lambda - \mathbb{E}\left[f^*\left(D(G(\mathbf{Z})) + \lambda\right)\right] \\
&= \min_{G \in \mathcal{G}} \max_{D \in \mathcal{F}, \lambda \in \mathbb{R}} \mathbb{E}_{P_{\mathbf{X}}}[D(\mathbf{X}) + \lambda] - \mathbb{E}\left[f^*\left(D(G(\mathbf{Z})) + \lambda\right)\right] \\
&= \min_{G \in \mathcal{G}} \max_{D \in \mathcal{F}} \mathbb{E}_{P_{\mathbf{X}}}[D(\mathbf{X})] - \mathbb{E}\left[f^*\left(D(G(\mathbf{Z}))\right)\right],
\end{aligned}
$$

which makes the proof complete. $\qquad\qquad\square$

## 4 Proof of Theorem 3

Theorem 3 is a direct application of the following lemma to Theorem 1 and Corollary 1.

**Lemma 4.** *Let $c$ be a lower semicontinuous non-negative cost function. Considering the $c$-transform operation $D^c$ defined in the text, the following holds for any continuous $D$*

$$
W_{cP}^*(D) = \mathbb{E}_P[D^c(\mathbf{X})]. \tag{10}
$$

*Proof.* We have

$$
\begin{aligned}
W_{cP}^*(D) &\overset{(a)}{=} \sup_Q \ \mathbb{E}_Q[D(\mathbf{X}')] - W_c(P, Q) \\
&\overset{(b)}{=} -\inf_Q \inf_{M \in \Pi(P,Q)} \mathbb{E}_M\left[c(\mathbf{X}, \mathbf{X}') - D(\mathbf{X}')\right] \\
&= -\inf_{Q, M \in \Pi(P,Q)} \mathbb{E}_M\left[c(\mathbf{X}, \mathbf{X}') - D(\mathbf{X}')\right] \\
&\overset{(c)}{\geq} -\mathbb{E}_P\left[\inf_{\mathbf{x}'} c(\mathbf{X}, \mathbf{x}') - D(\mathbf{x}')\right] \\
&= \mathbb{E}_P\left[\sup_{\mathbf{x}'} D(\mathbf{x}') - c(\mathbf{X}, \mathbf{x}')\right] \\
&\overset{(d)}{=} \mathbb{E}_P[D^c(\mathbf{X})].
\end{aligned}
$$

Here (a), (b), (d) hold according to the definitions. Moreover, we show (c) will hold with equality under the lemma's assumptions. $c(\mathbf{x}, \mathbf{x}') - D(\mathbf{x}')$ is lower semicontinuous, and hence for every $\epsilon > 0$ there exists a measurable function $v(\mathbf{x})$ such that for the coupling $M = \pi_{\mathbf{X}, v(\mathbf{X})}$ the absolute difference $\left|\mathbb{E}_M\left[c(\mathbf{X}, \mathbf{X}') - D(\mathbf{X}')\right] - \mathbb{E}_P\left[\inf_{\mathbf{x}'} c(\mathbf{X}, \mathbf{x}') - D(\mathbf{x}')\right]\right| < \epsilon$ is $\epsilon$-bounded. Therefore, $(c)$ holds with equality and the proof is complete. $\qquad\square$

## 5 Proof of Theorem 4

Consider a convex combination of functions from $\mathcal{F}_{nn}$ as $f_\alpha(\mathbf{x}) = \int \alpha(\mathbf{w}) f_{\mathbf{w}}(\mathbf{x}) \, d\mathbf{w}$ where $\alpha$ can be considered as a probability density function over feasible set $\mathcal{W}$. Consider $m$ samples $(\mathbf{W}_i)_{i=1}^m$ taken i.i.d. from $\alpha$. Since any $f_{\mathbf{w}}$ is $M$-bounded, according to Hoeffding's inequality for a fixed $\mathbf{x}$ we have

$$
\Pr\left(\left|\frac{1}{m}\sum_{i=1}^m f_{\mathbf{W}_i}(\mathbf{x}) - \mathbb{E}_{\mathbf{W} \sim \alpha}\left[f_{\mathbf{w}}(\mathbf{x})\right]\right| \geq \frac{\epsilon}{2}\right) \leq 2\exp\left(-\frac{m\epsilon^2}{8M^2}\right). \tag{11}
$$

Next we consider a $\delta$-covering for the ball $\{\mathbf{x} : \|\mathbf{x}\|_2 \leq R\}$, where we choose $\delta = \frac{\epsilon}{4L}$. We know a $\delta$-covering $\{\mathbf{x}_j : 1 \leq j \leq N\}$ exists with a bounded size $N \leq (12LR/\epsilon)^k$ [5]. Then, an application of the union bound implies

$$
\Pr\left(\max_{1 \leq j \leq N}\left|\frac{1}{m}\sum_{i=1}^m f_{\mathbf{W}_i}(\mathbf{x}_j) - \mathbb{E}_{\mathbf{W} \sim \alpha}\left[f_{\mathbf{w}}(\mathbf{x}_j)\right]\right| \geq \frac{\epsilon}{2}\right) \leq 2N\exp\left(-\frac{m\epsilon^2}{8M^2}\right)
$$

$$
\leq \exp\left(-\frac{m\epsilon^2}{8M^2} + k\log\left(\frac{12LR}{\epsilon}\right) + \log 2\right)
$$

Hence if we have $-\frac{m\epsilon^2}{8M^2} + k\log(\frac{12LR}{\epsilon}) + \log 2 < 0$ the above upper-bound is strictly less than 1, showing there exists at least one outcome $(\mathbf{w}_i)_{i=1}^m$ satisfying

$$\max_{1 \le j \le N} \left| \frac{1}{m} \sum_{i=1}^m f_{\mathbf{w}_i}(\mathbf{x}_j) - \mathbb{E}_{\mathbf{W} \sim \alpha}\big[ f_{\mathbf{W}}(\mathbf{x}_j) \big] \right| < \frac{\epsilon}{2}. \tag{12}$$

Then, we claim the following holds over the norm-bounded $\{\mathbf{x} : ||\mathbf{x}||_2 \le R\}$:

$$\sup_{||\mathbf{x}||_2 \le R} \left| \frac{1}{m} \sum_{i=1}^m f_{\mathbf{w}_i}(\mathbf{x}_j) - \mathbb{E}_{\mathbf{W} \sim \alpha}\big[ f_{\mathbf{W}}(\mathbf{x}_j) \big] \right| < \epsilon. \tag{13}$$

This is because due to the definition of a $\delta$-covering for any $||\mathbf{x}||_2 \le R$ there exists $\mathbf{x}_j$ for which $||\mathbf{x}_j - \mathbf{x}|| \le \frac{\epsilon}{4L}$. Then, since any $f_{\mathbf{w}}$ is supposed to be $L$-Lipschitz we have

$$\left| \frac{1}{m} \sum_{i=1}^m f_{\mathbf{w}_i}(\mathbf{x}_j) - \frac{1}{m} \sum_{i=1}^m f_{\mathbf{w}_i}(\mathbf{x}) \right| \le \frac{\epsilon}{4}, \quad \left| \mathbb{E}_{\mathbf{W} \sim \alpha}\big[ f_{\mathbf{W}}(\mathbf{x}_j) \big] - \mathbb{E}_{\mathbf{W} \sim \alpha}\big[ f_{\mathbf{W}}(\mathbf{x}) \big] \right| \le \frac{\epsilon}{4} \tag{14}$$

which together with (12) shows (13). Hence, if we choose

$$m = \frac{8M^2}{\epsilon^2} \big( k \log(12LR/\epsilon) + \log 2 \big) = \mathcal{O}\big( \frac{M^2 k \log(LR/\epsilon)}{\epsilon^2} \big) \tag{15}$$

there will be some weight assignments $(\mathbf{w}_i)_{i=1}^m$ such that their uniform combination $\frac{1}{m} \sum_{i=1}^m f_{\mathbf{w}_i}(\mathbf{x})$ $\epsilon$-approximates the convex combination $f_\alpha$ uniformly over $\{\mathbf{x} : ||\mathbf{x}||_2 \le R\}$.

## 6 Proof of Theorem 5

We show that for any distributions $P_0, P_1, P_2$ the following holds
$$\left| d_{f,W_1}(P_0, P_2) - d_{f,W_1}(P_1, P_2) \right| \le W_1(P_0, P_1). \tag{16}$$
The above inequality holds since if $Q_0$ and $Q_1$ solve the minimum sum optimization problems for $d_{f,W_1}(P_0, P_2), d_{f,W_1}(P_1, P_2)$, we have
$$d_{f,W_1}(P_0, P_2) - d_{f,W_1}(P_1, P_2) \le W_1(P_0, Q_1) - W_1(P_1, Q_1) \le W_1(P_0, P_1),$$
$$d_{f,W_1}(P_1, P_2) - d_{f,W_1}(P_0, P_2) \le W_1(P_1, Q_0) - W_1(P_0, Q_0) \le W_1(P_0, P_1)$$
where the second inequalities in both these lines follow from the symmetricity and triangle inequality property of the $W_1$-distance. Therefore, the following holds for any $Q$:
$$\left| d_{f,W_1}(P_{G_{\boldsymbol{\theta}}(\mathbf{Z})}, Q) - d_{f,W_1}(P_{G_{\boldsymbol{\theta}'}(\mathbf{Z})}, Q) \right| \le W_1(P_{G_{\boldsymbol{\theta}}(\mathbf{Z})}, P_{G_{\boldsymbol{\theta}'}(\mathbf{Z})}).$$
Hence, we only need to show $W_1(P_{G_{\boldsymbol{\theta}}(\mathbf{Z})}, Q)$ is changing continuously with $\boldsymbol{\theta}$ and is almost everywhere differentiable. We prove these things using a similar proof to [6]'s proof for the continuity of the first-order Wasserstein distance.

Consider two functions $G_{\boldsymbol{\theta}}, G_{\boldsymbol{\theta}'}$. The joint distribution $M$ for $(G_{\boldsymbol{\theta}}(\mathbf{Z}), G_{\boldsymbol{\theta}'}(\mathbf{Z}))$ is contained in $\Pi(P_{G_{\boldsymbol{\theta}}(\mathbf{Z})}, P_{G_{\boldsymbol{\theta}'}(\mathbf{Z})))$, which results in
$$W_1\big( P_{G_{\boldsymbol{\theta}}(\mathbf{Z})}, P_{G_{\boldsymbol{\theta}'}(\mathbf{Z})} \big) \le \mathbb{E}_M[||\mathbf{X} - \mathbf{X}'||]$$
$$= \mathbb{E}\big[ || G_{\boldsymbol{\theta}}(\mathbf{Z}) - G_{\boldsymbol{\theta}'}(\mathbf{Z}) || \big]. \tag{17}$$
If we let $\boldsymbol{\theta}' \to \boldsymbol{\theta}$ then $G_{\boldsymbol{\theta}}(\mathbf{z}) \to G_{\boldsymbol{\theta}'}(\mathbf{z})$ and hence $|| G_{\boldsymbol{\theta}'}(\mathbf{z}) - G_{\boldsymbol{\theta}}(\mathbf{z}) || \to 0$ hold pointwise. Since $\mathcal{X}$ is assumed to be compact, there exists some finite $R$ for which $0 \le ||\mathbf{x} - \mathbf{x}'|| \le R$ holds over the compact $\mathcal{X} \times \mathcal{X}$. Then the bounded convergence theorem implies $\mathbb{E}\big[ || G_{\boldsymbol{\theta}}(\mathbf{Z}) - G_{\boldsymbol{\theta}'}(\mathbf{Z}) || \big]$ converges to 0 as $\boldsymbol{\theta}' \to \boldsymbol{\theta}$. Then, since $W_1$-distance always takes non-negative values
$$W_1\big( P_{G_{\boldsymbol{\theta}}(\mathbf{Z})}, P_{G_{\boldsymbol{\theta}'}(\mathbf{Z})} \big) \xrightarrow{\boldsymbol{\theta}' \to \boldsymbol{\theta}} 0.$$
Thus, $W_1$ satisfies the discussed continuity property and as a result $d_{f,W_1}(P_{G_{\boldsymbol{\theta}}(\mathbf{Z})}, Q)$ changes continuously with $\boldsymbol{\theta}$. Furthermore, if $G_{\boldsymbol{\theta}}$ is locally-Lipschitz and its Lipschitz constant w.r.t. parameters $\boldsymbol{\theta}$ is bounded above by $L$,
$$d_{f,W_1}\big( P_{G_{\boldsymbol{\theta}}(\mathbf{Z})}, P_{G_{\boldsymbol{\theta}'}(\mathbf{Z})} \big) \le W_1\big( P_{G_{\boldsymbol{\theta}}(\mathbf{Z})}, P_{G_{\boldsymbol{\theta}'}(\mathbf{Z})} \big)$$
$$\le \mathbb{E}\big[ || G_{\boldsymbol{\theta}}(\mathbf{Z}) - G_{\boldsymbol{\theta}'}(\mathbf{Z}) || \big]$$
$$\le L||\boldsymbol{\theta} - \boldsymbol{\theta}'||, \tag{18}$$
which implies both $W_1(P_{G_{\boldsymbol{\theta}}(\mathbf{Z})}, Q)$ and $d_{f,W_1}(P_{G_{\boldsymbol{\theta}}(\mathbf{Z})}, Q)$ are everywhere continuous and almost everywhere differentiable w.r.t. $\boldsymbol{\theta}$.

# 7 Proof of Theorem 6

We first generalize the definition of the hybrid divergence to a general minimum-sum hybrid of an f-divergence and an optimal transport cost. For f-divergence $d_f$ and optimal transport cost $W_c$ corresponding to convex function $f$ and cost $c$ respectively, we define the following hybrid $d_{f,c}$ of the two divergence measures:

$$d_{f,c}(P_1, P_2) := \inf_Q W_c(P_1, Q) + d_f(Q, P_2). \tag{19}$$

**Lemma 5.** *Given a symmetric f-divergence $d_f$ with convex lower semicontinuous $f$ and a non-negative lower semicontinuous $c$, $d_{f,c}(P_1, P_2)$ will be a convex function of $P_1$ and $P_2$, and further satisfies the following generalization of the Kantorovich duality [7]:*

$$d_{f,c}(P_1, P_2) = \sup_{D \text{ c-concave}} \mathbb{E}_{P_1}[D(\mathbf{X})] - \mathbb{E}_{P_2}[f^*(D^c(\mathbf{X}))]. \tag{20}$$

*Proof.* According to the Kantorovich duality [7] we have

$$
\begin{aligned}
d_{f,c}(P_1, P_2) &\overset{(a)}{=} \inf_Q W_c(P_1, Q) + d_f(Q, P_2) \\
&\overset{(b)}{=} \inf_Q \sup_{D \text{ c-concave}} \mathbb{E}_{P_1}[D(\mathbf{X})] - \mathbb{E}_Q[D^c(\mathbf{X})] + d_f(Q, P_2) \\
&\overset{(c)}{=} \inf_Q \sup_{D \text{ c-concave}} \mathbb{E}_{P_1}[D(\mathbf{X})] - \mathbb{E}_Q[D^c(\mathbf{X})] + d_f(P_2, Q) \\
&\overset{(d)}{=} \sup_{D \text{ c-concave}} \inf_Q \mathbb{E}_{P_1}[D(\mathbf{X})] - \mathbb{E}_Q[D^c(\mathbf{X})] + d_f(P_2, Q) \\
&= \sup_{D \text{ c-concave}} \mathbb{E}_{P_1}[D(\mathbf{X})] + \inf_Q d_f(P_2, Q) - \mathbb{E}_Q[D^c(\mathbf{X})] \\
&\overset{(e)}{=} \sup_{D \text{ c-concave}} \mathbb{E}_{P_1}[D(\mathbf{X})] - d_f{}^*_{P_2}(D^c) \\
&\overset{(f)}{=} \sup_{D \text{ c-concave}} \mathbb{E}_{P_1}[D(\mathbf{X})] + \max_{\lambda \in \mathbb{R}} \lambda - \mathbb{E}_{P_2}[f^*(D^c(\mathbf{X}) + \lambda)] \\
&= \sup_{D \text{ c-concave}, \lambda \in \mathbb{R}} \mathbb{E}_{P_1}[D(\mathbf{X}) + \lambda] - \mathbb{E}_{P_2}[f^*(D^c(\mathbf{X}) + \lambda)]. \\
&= \sup_{D \text{ c-concave}} \mathbb{E}_{P_1}[D(\mathbf{X})] - \mathbb{E}_{P_2}[f^*(D^c(\mathbf{X}))].
\end{aligned}
$$

Here (a) holds according to the definition. (b) is a consequence of the Kantorovich duality ([7], Theorem 5.10). (c) holds becuase $d_f$ is assumed to be symmetric. (d) holds due to the generalized minimax theorem [3], since the space of distributions over compact $\mathcal{X}$ is convex and weakly compact, the set of c-concave functions is convex, the minimax objective is concave in $D$ and convex in $Q$. (e) holds according to the conjugate $d_P^*$'s definition, and (f) is based on our earlier result in (6). Note that the final expression is maximizing an objective linear in $P_2$, which is convex in $P_2$. The last equality holds since for any constant $\lambda \in \mathbb{R}$ if $D^c$ is the c-transform of $D$, $D^c + \lambda$ will be the c-transform of $D + \lambda$. Finally, note that $d_{f,c}(P_1, P_2)$ is the supremum of some linear functions of $P_1$ and $P_2$ with compact support sets. Hence $d_{f,c}$ will be a convex function of $P_1$ and $P_2$. □

Now we prove the following generalization of Theorem 6, which directly results in Theorem 6 for the difference norm cost $c_1(\mathbf{x}, \mathbf{x}') = \|\mathbf{x} - \mathbf{x}'\|$. Here note that for cost $c_1$ the c-transform of a 1-Lipschitz function $D$ will be $D$ itself, which implies if $f^* \circ D$ is 1-Lipschitz then

$$-f^*(D(G(\mathbf{Z}))) = \inf_{\mathbf{x}'} -f^*(D(\mathbf{x}')) + c_1(G(\mathbf{Z}), \mathbf{x}').$$

**Theorem** (Generalization of Theorem 6). *Assume $d_f$ is a symmetric f-divergence, i.e. $d_f(P, Q) = d_f(Q, P)$, satisfying the assumptions in Lemma 2. Suppose $\mathcal{F}$ is a convex set of continuous functions closed to constant additions and cost function $c$ is non-negative and continuous. Then, the minimax problem in Theorem 1 and Corollary 1 for the mixed divergence $d_{f,c}$ reduces to*

$$\min_{G \in \mathcal{G}} \max_{D \in \mathcal{F}} \mathbb{E}_{P_{\mathbf{X}}}[D(\mathbf{X})] + \mathbb{E}\big[\inf_{\mathbf{x}'} -f^*(D(\mathbf{x}')) + c(G(\mathbf{Z}), \mathbf{x}')\big]. \tag{21}$$

*Proof.* Accoriding to Lemma 5, $d_{f,c}(P, Q)$ satisfies the convexity property in $Q$. Hence, the assumptions of Theorem 1 and Corollary 1 hold and we only need to plug in the conjugate $d_{f,c\,P_1}^*$ into Corollary 1. According to the definition,

$$
\begin{aligned}
d_{f,c\,P_1}^*(D) &= \sup_{P_2} \mathbb{E}_{P_2}[D(\mathbf{X})] - d_{f,c}(P_1, P_2) \\
&= \sup_{P_2} \sup_{Q} -W_c(P_1, Q) - d_f(Q, P_2) + \mathbb{E}_{P_2}[D(\mathbf{X})] \\
&= \sup_{Q} \sup_{P_2} -W_c(P_1, Q) - d_f(Q, P_2) + \mathbb{E}_{P_2}[D(\mathbf{X})] \\
&= \sup_{Q} -W_c(P_1, Q) + \sup_{P_2} \mathbb{E}_{P_2}[D(\mathbf{X})] - d_f(Q, P_2) \\
&= \sup_{Q} -W_c(P_1, Q) + d_{f\,Q}^*(D) \\
&\overset{(g)}{=} \sup_{Q} -W_c(P_1, Q) + \min_{\lambda \in \mathbb{R}} -\lambda + \mathbb{E}_Q[f^*(D(\mathbf{X}) + \lambda)] \\
&= \sup_{Q} \min_{\lambda \in \mathbb{R}} -W_c(P_1, Q) - \lambda + \mathbb{E}_Q[f^*(D(\mathbf{X}) + \lambda)] \\
&\overset{(h)}{=} \min_{\lambda \in \mathbb{R}} \sup_{Q} -W_c(P_1, Q) - \lambda + \mathbb{E}_Q[f^*(D(\mathbf{X}) + \lambda)] \\
&\overset{(i)}{=} \inf_{\lambda \in \mathbb{R}} -\lambda + \mathbb{E}_{P_1}\big[\big(f^* \circ (D + \lambda)\big)^c(\mathbf{X})\big].
\end{aligned}
$$

Here (g) holds based on our earlier result in (6). (h) is a consequence of the minimax theorem, since the space of distributions over compact $\mathcal{X}$ is convex and compact, and the objective is concave in $\lambda$ and lower semicontinuous and convex in $Q$. (i) is implied by Lemma 3. Therefore, according to Corollary 1

$$
\begin{aligned}
&\min_{P_{G(\mathbf{Z})} \in \mathcal{P}_\mathcal{G}} \min_{Q_\mathbf{X}} d_{f,c}(P_{G(\mathbf{Z})}, Q) + \max_{D \in \mathcal{F}} \big\{ \mathbb{E}_{P_\mathbf{X}}[D(\mathbf{X})] - \mathbb{E}_Q[D(\mathbf{X})] \big\} \\
&= \min_{G \in \mathcal{G}} \max_{D \in \mathcal{F}} \mathbb{E}_{P_\mathbf{X}}[D(\mathbf{X})] - d_{f,c\,P_{G(\mathbf{Z})}}^*(D) \\
&= \min_{G \in \mathcal{G}} \max_{D \in \mathcal{F}} \mathbb{E}_{P_\mathbf{X}}[D(\mathbf{X})] + \max_{\lambda \in \mathbb{R}} \lambda - \mathbb{E}\big[\big(f^* \circ (D + \lambda)\big)^c(G(\mathbf{Z}))\big] \\
&= \min_{G \in \mathcal{G}} \max_{D \in \mathcal{F}, \lambda \in \mathbb{R}} \mathbb{E}_{P_\mathbf{X}}[D(\mathbf{X}) + \lambda] - \mathbb{E}\big[\big(f^* \circ (D + \lambda)\big)^c(G(\mathbf{Z}))\big] \\
&\overset{(j)}{=} \min_{G \in \mathcal{G}} \max_{D \in \mathcal{F}} \mathbb{E}_{P_\mathbf{X}}[D(\mathbf{X})] - \mathbb{E}\big[(f^* \circ D)^c(G(\mathbf{Z}))\big] \\
&= \min_{G \in \mathcal{G}} \max_{D \in \mathcal{F}} \mathbb{E}_{P_\mathbf{X}}[D(\mathbf{X})] - \mathbb{E}\big[\sup_{\mathbf{x}'} f^*(D(\mathbf{x}')) - c\big(G(\mathbf{Z}), \mathbf{x}'\big)\big] \\
&= \min_{G \in \mathcal{G}} \max_{D \in \mathcal{F}} \mathbb{E}_{P_\mathbf{X}}[D(\mathbf{X})] + \mathbb{E}\big[\inf_{\mathbf{x}'} -f^*(D(\mathbf{x}')) + c\big(G(\mathbf{Z}), \mathbf{x}'\big)\big].
\end{aligned}
$$

Here (j) holds since $\mathcal{F}$ is assumed to be closed to constant additions. Hence, the proof is complete. $\quad\square$

## 8 Proof of Theorem 7

Consider distributions $P_0, P_1, P_2$. Let $Q_0, Q_1$ be the optimal solutions to the minimum sum optimization problems for $d_{f,W_2}(P_0, P_2)$ and $d_{f,W_2}(P_1, P_2)$, respectively. Then, according to the definition

$$
\begin{aligned}
d_{f,W_2}(P_0, P_2) - d_{f,W_2}(P_1, P_2) &\le W_2^2(P_0, Q_1) - W_2^2(P_1, Q_1), \\
d_{f,W_2}(P_1, P_2) - d_{f,W_2}(P_0, P_2) &\le W_2^2(P_1, Q_0) - W_2^2(P_0, Q_0)
\end{aligned}
$$

which implies

$$
\big| d_{f,W_2}(P_0, P_2) - d_{f,W_2}(P_1, P_2) \big| \le \sup_{Q} \big| W_2^2(P_0, Q) - W_2^2(P_1, Q) \big|.
$$

Hence, for $G_{\boldsymbol{\theta}}$, $G_{\boldsymbol{\theta}'}$ and any distribution $P_2$ we have

$$
\big| d_{f,W_2}(P_{G_{\boldsymbol{\theta}}(\mathbf{Z})}, P_2) - d_{f,W_2}(P_{G_{\boldsymbol{\theta}'}(\mathbf{Z})}, P_2) \big| \le \sup_{Q} \big| W_2^2(P_{G_{\boldsymbol{\theta}}(\mathbf{Z})}, Q) - W_2^2(P_{G_{\boldsymbol{\theta}'}(\mathbf{Z})}, Q) \big|. \quad (22)
$$

Fix a distribution $Q$ over the compact $\mathcal{X}$. Then, for any $(G_{\boldsymbol{\theta}}(\mathbf{Z}), \mathbf{X}')$ whose joint distribution is in $\Pi(P_{G_{\boldsymbol{\theta}}(\mathbf{Z})}, Q)$, $(G_{\boldsymbol{\theta}'}(\mathbf{Z}), \mathbf{X}')$ has a joint distribution in $\Pi(P_{G_{\boldsymbol{\theta}'}(\mathbf{Z})}, Q)$. Moreover, since $\mathcal{X}$ is a compact set in a Hilbert space, any $\mathbf{x} \in \mathcal{X}$ is norm-bounded for some finite $R$ as $\|\mathbf{x}\| \leq R$, which implies

$$\left| W_2^2(P_{G_{\boldsymbol{\theta}}(\mathbf{Z})}, Q) - W_2^2(P_{G_{\boldsymbol{\theta}'}(\mathbf{Z})}, Q) \right|$$

$$\leq \sup_{M_{\mathbf{Z},\mathbf{X}'} \in \Pi(P_{\mathbf{Z}}, Q)} \left| \mathbb{E}_M \left[ \| G_{\boldsymbol{\theta}}(\mathbf{Z}) - \mathbf{X}' \|^2 - \| G_{\boldsymbol{\theta}'}(\mathbf{Z}) - \mathbf{X}' \|^2 \right] \right|$$

$$\leq \sup_{M_{\mathbf{Z},\mathbf{X}'} \in \Pi(P_{\mathbf{Z}}, Q)} \mathbb{E}_M \left[ \left| \|G_{\boldsymbol{\theta}}(\mathbf{Z})\|^2 - \|G_{\boldsymbol{\theta}'}(\mathbf{Z})\|^2 \right| + 2\|\mathbf{X}'\| \, \| G_{\boldsymbol{\theta}'}(\mathbf{Z}) - G_{\boldsymbol{\theta}}(\mathbf{Z}) \| \right]$$

$$\leq \mathbb{E}_{P_{\mathbf{Z}}} \left[ \left| \|G_{\boldsymbol{\theta}}(\mathbf{Z})\|^2 - \|G_{\boldsymbol{\theta}'}(\mathbf{Z})\|^2 \right| + 2R \, \| G_{\boldsymbol{\theta}'}(\mathbf{Z}) - G_{\boldsymbol{\theta}}(\mathbf{Z}) \| \right].$$

Taking a supremum over $Q$ from both sides of the above inequality shows

$$\sup_Q \left| W_2^2(P_{G_{\boldsymbol{\theta}}(\mathbf{Z})}, Q) - W_2^2(P_{G_{\boldsymbol{\theta}'}(\mathbf{Z})}, Q) \right|$$

$$\leq \mathbb{E}_{P_{\mathbf{Z}}} \left[ \left| \|G_{\boldsymbol{\theta}}(\mathbf{Z})\|^2 - \|G_{\boldsymbol{\theta}'}(\mathbf{Z})\|^2 \right| + 2R \, \| G_{\boldsymbol{\theta}'}(\mathbf{Z}) - G_{\boldsymbol{\theta}}(\mathbf{Z}) \| \right]. \tag{23}$$

Since $G_{\boldsymbol{\theta}}$ changes continuously with $\boldsymbol{\theta}$, $\left| \|G_{\boldsymbol{\theta}}(\mathbf{z})\|^2 - \|G_{\boldsymbol{\theta}'}(\mathbf{z})\|^2 \right| + 2R \, \| G_{\boldsymbol{\theta}'}(\mathbf{z}) - G_{\boldsymbol{\theta}}(\mathbf{z}) \| \to 0$ as $\boldsymbol{\theta}' \to \boldsymbol{\theta}$ holds pointwise. Therefore, since $\mathcal{X}$ is compact and hence bounded, the bounded convergence theorem together with (23) implies

$$\sup_Q \left| W_2^2(P_{G_{\boldsymbol{\theta}}(\mathbf{Z})}, Q) - W_2^2(P_{G_{\boldsymbol{\theta}'}(\mathbf{Z})}, Q) \right| \xrightarrow{\boldsymbol{\theta}' \to \boldsymbol{\theta}} 0. \tag{24}$$

Now, combining (22) and (24) shows for any distribution $P_2$

$$\left| d_{f,W_2}(P_{G_{\boldsymbol{\theta}}(\mathbf{Z})}, P_2) - d_{f,W_2}(P_{G_{\boldsymbol{\theta}'}(\mathbf{Z})}, P_2) \right| \xrightarrow{\boldsymbol{\theta}' \to \boldsymbol{\theta}} 0. \tag{25}$$

Also, if we further assume $G_{\boldsymbol{\theta}}$ is bounded by $T$ locally-Lipschitz w.r.t. $\boldsymbol{\theta}$ with Lipschitz constant $L$, then

$$\sup_Q \left| W_2^2(P_{G_{\boldsymbol{\theta}}(\mathbf{Z})}, Q) - W_2^2(P_{G_{\boldsymbol{\theta}'}(\mathbf{Z})}, Q) \right|$$

$$\leq \mathbb{E}_{P_{\mathbf{Z}}} \left[ \left| \|G_{\boldsymbol{\theta}}(\mathbf{Z})\|^2 - \|G_{\boldsymbol{\theta}'}(\mathbf{Z})\|^2 \right| + 2R \, \| G_{\boldsymbol{\theta}'}(\mathbf{Z}) - G_{\boldsymbol{\theta}}(\mathbf{Z}) \| \right] \tag{26}$$

$$\leq \mathbb{E}_{P_{\mathbf{Z}}} \left[ \left| (\|G_{\boldsymbol{\theta}}(\mathbf{Z})\| + \|G_{\boldsymbol{\theta}'}(\mathbf{Z})\|) (\|G_{\boldsymbol{\theta}}(\mathbf{Z})\| - \|G_{\boldsymbol{\theta}'}(\mathbf{Z})\|) \right| + 2R \, \| G_{\boldsymbol{\theta}'}(\mathbf{Z}) - G_{\boldsymbol{\theta}}(\mathbf{Z}) \| \right]$$

$$\leq \mathbb{E}_{P_{\mathbf{Z}}} \left[ 2T \left| \|G_{\boldsymbol{\theta}}(\mathbf{Z})\| - \|G_{\boldsymbol{\theta}'}(\mathbf{Z})\| \right| + 2R \, \| G_{\boldsymbol{\theta}'}(\mathbf{Z}) - G_{\boldsymbol{\theta}}(\mathbf{Z}) \| \right]$$

$$\leq \mathbb{E}_{P_{\mathbf{Z}}} \left[ 2(T + R) \, \| G_{\boldsymbol{\theta}'}(\mathbf{Z}) - G_{\boldsymbol{\theta}}(\mathbf{Z}) \| \right]$$

$$\leq 2(T + R)L \, \| \boldsymbol{\theta}' - \boldsymbol{\theta} \|,$$

implying $d_{f,W_2}(P_{G_{\boldsymbol{\theta}}(\mathbf{Z})}, Q)$ is continuous everywhere and differentiable almost everywhere as a function of $\boldsymbol{\theta}$.

## 9 Proof of Theorem 8

Note that applying the generalized version of Theorem 6 proved in the Appendix to difference norm-squared cost $c_2(\mathbf{x}, \mathbf{x}') = \|\mathbf{x} - \mathbf{x}'\|^2$ reveals that for a symmetric f-divergence $d_f$ and convex

set $\mathcal{F}$ closed to constant additions the minimax problem in Theorem 1 and Corollary 1 for the mixed divergence $d_{f,c_2}$ reduces to

$$\min_{G \in \mathcal{G}} \max_{D \in \mathcal{F}} \mathbb{E}_{P_\mathbf{X}}[D(\mathbf{X})] + \mathbb{E}\big[\min_{\mathbf{x}'} -f^*(D(\mathbf{x}')) + c_2\big(G(\mathbf{Z}),\mathbf{x}'\big)\big]$$

$$= \min_{G \in \mathcal{G}} \max_{D \in \mathcal{F}} \mathbb{E}_{P_\mathbf{X}}[D(\mathbf{X})] + \mathbb{E}\big[\min_{\mathbf{x}'} -f^*(D(\mathbf{x}')) + \|G(\mathbf{Z}) - \mathbf{x}'\|^2\big] \qquad (27)$$

$$= \min_{G \in \mathcal{G}} \max_{D \in \mathcal{F}} \mathbb{E}_{P_\mathbf{X}}[D(\mathbf{X})] + \mathbb{E}\big[\min_{\mathbf{u}} -f^*\big(D(G(\mathbf{Z}) + \mathbf{u})\big) + \|\mathbf{u}\|^2\big].$$

Here the last equality follows the change of variable $\mathbf{u} = \mathbf{x}' - G(\mathbf{Z})$. Also, note that $d_{f,W_2}$ defined in the main text is the same as the special case of the generalized hybrid divergence $d_{f,c}$ with cost $c_2$. Hence, the proof is complete.

## 10  Two additional examples for convex duality framework applied to Wasserstein distances

### 10.1  Total variation distance: Energy-based GAN

Consider the total variation distance $\delta(P,Q)$ which is defined as

$$\delta(P,Q) := \sup_{A \in \Sigma} \big|P(A) - Q(A)\big|, \qquad (28)$$

where $\Sigma$ is the set all Borel subsets of support set $\mathcal{X}$. More generally we consider $\delta_m(P,Q) = m\delta(P,Q)$ for any positive $m > 0$. Under mild assumptions, the total variation distance can be cast as a Wasserstein distance for the indicator cost $c_{m,I}(\mathbf{x},\mathbf{x}') = m\,\mathbb{I}(\mathbf{x} \neq \mathbf{x}')$ [7], i.e. $\delta_m(P,Q) = OT_{c_{m,I}}(P,Q)$. Note that $c_{m,I}$ is a lower semicontinuous distance function, and hence Lemma 3 applies to $c_{m,I}$ indicating

$$\delta_{m\,P}^*(D) = OT_{c_{I,m}\,P}^*(D)$$
$$= \mathbb{E}_P[D^{c_{I,m}}(\mathbf{X})]$$
$$= \mathbb{E}_P\big[\sup_{\mathbf{x}'} D(\mathbf{x}') - m\,c_I(\mathbf{X},\mathbf{x}')\big]$$
$$= \mathbb{E}_P\big[\max\big\{D(\mathbf{X}),\ \max_{\mathbf{x}'} D(\mathbf{x}') - m\big\}\big]$$
$$= \mathbb{E}_P\big[\max\big\{m + D(\mathbf{X}) - \max_{\mathbf{x}'} D(\mathbf{x}'),\ 0\big\}\big] + \max_{\mathbf{x}'} D(\mathbf{x}') - m$$

Without loss of generality, we can assume that the maximum discriminator output is always 0 which results in

$$\delta_{m\,P}^*(D) = \mathbb{E}_P\big[\max\big\{m + D(\mathbf{X}),\ 0\big\}\big] - m$$

Therefore, the minimax problem in Corollaries 1,2 for the total variation distance will be

$$\min_{G \in \mathcal{G}} \max_{D \in \mathcal{F}} \mathbb{E}_P[D(\mathbf{X})] - \delta_{m\,P}^*(D)$$
$$= \min_{G \in \mathcal{G}} \max_{D \in \mathcal{F}} \mathbb{E}_P[D(\mathbf{X})] - \mathbb{E}_P\big[\max\{m + D(G(\mathbf{Z})),\ 0\}\big] + m$$
$$= \min_{G \in \mathcal{G}} \max_{-D \in \mathcal{F}} -\mathbb{E}_P[D(\mathbf{X})] - \mathbb{E}_P\big[\max\{m - D(G(\mathbf{Z})),\ 0\}\big] + m$$
$$= \min_{G \in \mathcal{G}} \max_{\tilde{D} \in \mathcal{F}} -\mathbb{E}_P[\tilde{D}(\mathbf{X})] - \mathbb{E}_P\big[\max\{m - \tilde{D}(G(\mathbf{Z})),\ 0\}\big] + m$$

where the last equality follows from the assumption that for any $D \in \mathcal{F}$ we have $-D \in \mathcal{F}$. Since $D$ is assumed to be non-positive, $\tilde{D}$ takes non-negative values. Note that this problem is equivalent to a minimax game where discriminator $D$ is *minimizing* the following cost over $\mathcal{F}$:

$$L_D(G,D) = \mathbb{E}_P[D(\mathbf{X})] + \mathbb{E}_P\big[\max\big\{m - D(G(\mathbf{Z})),\ 0\big\}\big] \qquad (29)$$

which is also the discriminator cost function in the energy-based GAN [8]. Hence, for any fixed $G \in \mathcal{G}$, the optimal discriminator $D \in \mathcal{F}$ for the total variation's minimax problem is the same as the energy-based GAN's optimal discriminator.

## 10.2 Second-order Wasserstein distance: the LQG setting

Consider the second-order Wasserstein distance $W_2(P, Q)$, and suppose $\mathcal{F}$ is the set of quadratic functions over $\mathbf{X}$, which is a linear space. Also assume the generator $G$ is a linear function and the $r$-dimensional noise $\mathbf{Z}$ is Gaussianly-distributed with zero-mean and identity covariance matrix $I_{r \times r}$. According to the interpretation provided in Corollary 2, the second-order Wasserstein GAN finds the multivariate Gaussian distribution with rank $r$ covariance matrix minimizing the $W_2$ distance to the set of distributions with their second-order moments matched to $P_{\mathbf{X}}$'s moments.

Since the value of $\mathbb{E}[\|\mathbf{X} - G(\mathbf{Z})\|^2]$ depends only on the second-order moments of the vector $[\mathbf{X}, G(\mathbf{Z})]$, we can minimize the $W_2$-distance between the two sets by minimizing this expectation over Gaussianly-distributed vectors $[\mathbf{X}, G(\mathbf{Z})]$ subject to a rank $r$ covariance matrix for $[G(\mathbf{Z})]$ and a pre-determined covariance matrix for $[\mathbf{X}]$. Hence, the optimal $G^*$ simply corresponds to the $r$-PCA solution for $P_{\mathbf{X}}$.

This example shows Theorem 3 provides another way to recover [9]'s main result under the linear generator, quadratic discriminator and Gaussianly-distributed data assumptions.