[Reviews · NeurIPS 2018]

Reviewer 1



Summary: The paper provides a new convex duality framework to understand the training for GANs when the discriminator is constrained - which is a a relevant practical and useful departure from the theory in vanilla GAN proposed in Goodfellow et al 2014, where the treatment was if discriminator was allowed an unconstrained set of functions. Detailed Comments: Clarity and Originality : The paper is well written and the proposed framework of convex duality is quite novel and rigorously treated. It is an important contribution to NIPS this year. Quality and Significance : The problem is quite significant as in general discriminators are constrained to smaller classes such as neural nets which certainly change the JS divergence treatment as in Goodfellow et al 2014. They show that if the class F is convex, then the minimizing JS divergence for unconstrained class is equivalent to minimizing JS divergence between the real distribution and the closest within the generative class penalized to up to sharing of same moments. This is quite a neat formulation and perhaps one of its kind. This framework accounts not only for vanilla GAN but also for f-GAN and WGAN as shown in Section 4. While in general the neural nets may not be convex, in Section 5 the authors circumvent this by considering an approximate solution in the convex hull of neural nets, which also helps suggest using a uniform combination of multiple discriminator nets. The only lacunae which can be felt is lack of enough experimental validation. It will be nice to add further tests on more datasets. However, overall this is a sure accept paper.

Reviewer 2



So I generally like the subject and theory of this paper very much. The authors illuminate some of the connections between many different GAN formulations and regularizations under the same convex dual framework. Unfortunately, the overall message I think will be lost on most readers, as the main bullet points of the paper seem fairly disjoint and sometimes are unclear w.r.t. GANs in the literature. For instance, when the distribution Q is introduced, it’s unclear what this is and how it is optimized. It’s finally clarified a bit in equation (21), but sadly there’s no real experiments to demonstrate that this formulation is sensible. When we finally get to equation (20), it’s unclear what this form gets us (other than eqs 21 and 22). The biggest problem though is the lack of experimentation. I like the theory, but I’m constantly asking myself what is the consequences of all of these lemmas and whether any of this is grounded in experimentation. The authors provide some experiments on MNIST, but we’re to be led to believe that some samples might be better than others based on visualization alone (I certainly cannot tell that some sets are better than others or even good w.r.t. the target dataset). The reason we need more experimentation, particularly in this paper despite it’s contribution being mostly theoretical, is that the family of functions F is a central player in this paper and the claims being made. In neural network optimization, F is not constrained only by the architecture / regularization, but by gradient descent based optimization. In this form, I would not recommend the paper being accepted without a better balance of theory and experimentation showing that the theory is grounded as well as to provide much needed interpretation for the reader. I would strongly encourage the authors to restructure and condense some of the theory to allow for room for this. For instance, can we explore equations 21 and 22 more with toy data? Are there ways of illuminating for the reader the importance of the 3rd player (as realized through Q)? Are there experiments that can better show the importance of the mixture model w.r.t. the duality framework? =============== So, as I said I really like this work. The proposed changes I think sound marginally better, but what I'm hoping for is a more thorough analysis w.r.t. the method that goes beyond just inception scores (its unclear after reading the response that authors intend to do any more than this). I think for this work to have broader impact, that some additional analyses are necessary illuminating my the new objective is similar to / different from existing methods (WGAN, Roth et al., Mescheder et. al., Spectral norm, etc). So please consider including more analysis in the final version if the paper gets accepted.

Reviewer 3



This paper provides a convex duality framework for analyzing GANs, many previous works are just special cases under this framework. All the proofs are based on the function space for both the discriminator and the generator which are constrained to a convex set, but in reality, it should be based on parametric space and the optimization is not convex. To use the theorems proposed in this paper, the authors use a linear combination of neural network functions instead. In the experiments, they get good results when using a linear combination of the same already trained neural net. The authors should emphasize the theoretical results are valid on the functional space, but not in the parameter space. The experiments are rather weak, they should provide results of using various combinations of neural net functions and see how the results behave. line 106 framework typo